# Adequacy of Parenteral Nutrition in Preterm Infants According to Current Recommendations: A Study in A Spanish Hospital

**DOI:** 10.3390/ijerph17062131

**Published:** 2020-03-23

**Authors:** Ana María Sánchez-García, Ana Zaragoza-Martí, Ana Cristina Murcia-López, Andrés Navarro-Ruiz, Ana Noreña-Peña

**Affiliations:** 1Department of Pharmacy Services, University General Hospital of Elche, 03203 Elche, Spain; ana25790@gmail.com (A.M.S.G.); murcia_ana@gva.es (A.C.M.L.); navarro_and@gva.es (A.N.R.); 2Department of Nursing, Faculty of Health Science, University of Alicante, 03690 Alicante, Spain; ana.norena@ua.es

**Keywords:** preterm newborn, parenteral nutrition, protocols

## Abstract

Background: In preterm infants, it is important to ensure adequate nutritional intake to accomplish foetal growth requirements. This study evaluated clinical practice regarding the prescription of parenteral nutrition in preterm infants in the neonatology unit of a tertiary hospital. Methods: It was a retrospective observational study of a sample of preterm infants (n = 155) born between January 2015 and December 2017 who were prescribed parenteral nutrition. Compliance with the hospital’s protocol and with the guidelines of the scientific societies American Society for Parenteral and Enteral Nutrition (ASPEN), European Society for Clinical Nutrition and Metabolism (ESPEN) and Spanish Society of Clinical Nutrition and Metabolism (SENPE) was evaluated. The differences in macronutrient intake and total duration of parenteral nutrition were analysed according to gestational age and birth weight. Results: The established protocol was followed in a high percentage (95.5%–100%) except with respect to the initiation of supplying established trace elements (64.9%). Compliance with the recommendations set forth in the guidelines was between 82.1% and 100%, with the exception of the initial carbohydrate intake recommended by ASPEN and ESPEN, for which compliance was 8.3%. Lower gestational age and birth weight were correlated with longer duration of parenteral nutrition (*p* < 0.001). Conclusions: A lower gestational age and birth weight are related to a longer duration of parenteral nutrition. The results of this study demonstrate the importance of developing and evaluating protocols in clinical practice.

## 1. Introduction

Worldwide, prematurity is the leading cause of mortality in children under 5 years of age [1]. It is estimated that 15 million births (11%) occur before 37 weeks of gestation, and the incidence of preterm births is increasing [2]. In Europe, annual prematurity rates range from 5.2 to 10.4% [3] and are lower than rates in countries such as the United States (12%) [4]. In Spain, according to the latest data from the National Institute of Statistics, preterm births account for 6.4% of all births [5]. Although the incidence of preterm births is increasing, the survival rate of preterm infants, especially in developed countries, has been increasing in recent years due to advances in medical care, among which is the use of nutritional support [4].

The preterm population presents a high risk of postnatal malnutrition, and this risk is higher for infants of low gestational age (GA) because nutritional reserves are lower in younger preterm infants [6]. To avoid this risk, nutritional intervention should begin at birth [7]. Initiation of correct nutrient intake as soon as possible is related to a decrease in postnatal growth restriction; caloric intake exceeding baseline requirements can improve the growth rate even more, especially in the most preterm infants, whose nutritional needs are greater [8]. This contributes to preventing the consequences related to poor growth and to decreasing associated health and social costs, since poor growth in children is clearly associated with the risk of developing physical or neurological problems that require greater health and social care, leading to higher costs [9].

To ensure proper nutritional intervention, it is essential that neonatal units develop protocols based on recommendations that have been validated and agreed upon by scientific societies [10], such as the guidelines established by American Society for Parenteral and Enteral Nutrition (ASPEN) [11], European Society for Clinical Nutrition and Metabolism (ESPEN) [12] and Spanish Society of Clinical Nutrition and Metabolism (SENPE) [13]. These guidelines are important tools that have been validated by health professionals and that help improve the management of nutritional needs in different population groups.

The foregoing considerations show that it is important to assess whether the clinical practice conducted in nutritional interventions in the preterm population is adequate. Currently, published studies evaluating this clinical topic are scarce, and their results show that care practice differs from the recommendations established in the guidelines and that in most cases the recommended intake is not administered; there is also great variability in the prescription of these supplements in different neonatal care units, and nearly half of such units do not have established protocols [14,15,16,17,18]. Consequently, studies are needed that centre their objective in evaluating the degree to which hospital services comply with the established parenteral nutrition (PN) protocols for preterm infants and, in this way, ensure adequate nutrient intake and achieve proper nutritional monitoring from birth [19].

This study focuses on evaluating whether clinical practice regarding the prescription of PN in preterm infants in the neonatal unit of a tertiary hospital complies with established protocols and current published recommendations. In addition, it analyses the differences in macronutrient intake and total duration of PN as a function of GA and birth weight (BW).

## 2. Methods 

### 2.1. Design and Study Population

A retrospective observational study was performed in the pharmacy service of a tertiary hospital, General University Hospital of Elche (Spain), from January 2015 to December 2017. All preterm infants admitted to the neonatology unit (in which the neonatal intensive care unit is located) that were prescribed PN (n = 155), excluding those admitted to this unit who initiated PN in other hospitals, were included in the study. Prematurity was classified according to GA (extremely preterm if GA < 28 weeks, preterm if GA = 28–34 weeks and late preterm if GA = 34–37 weeks) and according to BW (≤ 1500 g or >1500 g).

#### 2.1.1. Variables

Among the variables collected were sociodemographic (sex, GA, BW), maternal history (maternal age, type of pregnancy (multiple or single) and number of pregnancies-abortions-live births (PAL)) and variables related to the medical prescription of PN (initial PN in the first 24 h of life, total days of PN, initial micronutrient intake on day 1 and maximum intake, prescription on day 1 of vitamins, zinc and calcium, and prescription on day 4 or 5 of trace elements). The PN variables related to water, electrolytes (potassium, sodium, chloride and phosphate) and magnesium intake were excluded due to their clinical variability (dependence on renal function, respiratory and cardiac complications, among others) and/or variability in analytical data of the newborn.

#### 2.1.2. Procedure

The data referring to sociodemographic variables and maternal history were collected by a member of the research team through the hospital management software Mizar®. This is electronic programme used and developed in Valencian Community. For data related to prescribed PN intake, the manual prescriptions made by the paediatrician and sent to the pharmacy service and the information in the HospiWin® 2000 v8 programme used for formulation of PN by the responsible pharmacist were reviewed. The pharmacist enters the nutritional intakes prescribed by the doctor and the data of the newborn as the weight in the Hospiwin program. This program calculates the total nutrient intake that must be prepared to administer to the newborn.

#### 2.1.3. Ethical Considerations

The present study was conducted according to the ethical standards of biomedical research and data protection set forth in the Declaration of Helsinki. This study was evaluated and approved by the Research Ethics Committee of the General University Hospital of Elche in May 2018 (code PI 17/2018). To ensure data confidentiality, patient identification was coded.

#### 2.1.4. Statistical Analysis

The statistical package IBM SPSS Statistics v.23.0 (IBM Company, Spain) for Windows was used to perform the statistical analysis. The categorical variables were analysed using frequencies; the quantitative variables are expressed as the mean (± SD) when the values follow a normal distribution and as the median (± IQR) when the values not follow a normal distribution. The X2 test was used to compare categorical data, and the Kruskal–Wallis and Mann–Whitney U tests were used to compare quantitative data, assigning a significance level of 0.05. Spearman’s rho coefficient was used to determine the association between total days of PN and GA and BW. 

## 3. Results

Table 1 shows the distribution of sociodemographic and maternal history variables. A total of 52.8% of the preterm infants were male, and there were no statistically significant differences with respect to the types of preterm birth. The mean GA was 30.0 (± 3.0) weeks, and the median BW was 1200 (±558) grams. Single-foetus pregnancies accounted for 67.7% of the cases, with no differences between groups. Regarding maternal history, the median maternal age was 33.0 (± 9.0) years, with no statistically significant differences between groups. A total of 66.2% of the mothers had been pregnant before and the 47.7% had had abortions, and 60.3% had more than one lived child; none of these parameters differed significantly among the groups, although a higher percentage of abortions was observed in the extremely preterm group (61.8%).

The general distribution of prescribed nutritional doses of nutrients can be seen in Table 2. In 84% of preterm infants who received PN, PN was prescribed during the first day of life; the median intake of protein, lipids and carbohydrates was 2.0 (±0.5), 1.0 (±0.5) and 7.0 (±0.5) g/kg/day, respectively. The median maximum macronutrient intake was 3.0 (±0.2) g/kg/day in the case of proteins, 3.0 (±0.5) g/kg/day in the case of lipids and 11.0 (±2.5) g/kg/day in the case of carbohydrates. In only one case (0.6%) was the initiation of calcium and zinc not prescribed on the first day; 97.4% of the infants were prescribed vitamins on the first day. On the other hand, 35.1% did not receive trace elements on days 4 or 5, although of these, 86.7% initiated trace element intake on day 6.

A comparison of the PN prescriptions with the recommendations of scientific societies (SENPE, ESPEN and ASPEN) and with the recommendations established in the hospital protocol is shown in Table 3. The results show that the macronutrient intakes were adjusted to the established hospital protocol (compliance was 95.5% to 100%), as was the initiation of calcium (99.4%), zinc (99.4%) and vitamins (97.4%) However, as indicated, in preterm infants who continued with PN for more than 4 days, the initiation of trace elements on days 4 or 5 was not frequently prescribed (35.1%), with compliance in this case of 64.9%.

Concerning compliance with the guidelines, there were differences in the recommendations regarding macronutrients. The initial carbohydrate intake recommended in the SENPE 2017 and ASPEN 2002 guidelines is higher than that recommended in the ESPEN 2005 and hospital protocol, so that our prescriptions adjusted by 8.3%. Compliance with the remaining recommendations for macronutrients was between 82.1% and 100%, with no significant differences among them. In this case, the compliance with recommended levels of the studied micronutrients could not be evaluated because these guidelines do not specify when micronutrient supplementation should be initiated. The only specification is for initiation of vitamins on the first day of life, and this is found only in the ESPEN guidelines. In this study, compliance with this recommendation was 97.4%.

The analysis of macronutrient intake and total days of PN as a function of the type of prematurity according to GA is detailed in Table 4. The results show that there were statistically significant differences within groups in the prescription of maximum lipid intake and the initiation of carbohydrates; these were lower and higher, respectively, in the case of late preterm infants. With respect to the other supplements, there were no differences among the groups. On the other hand, there were significant differences in the total days of PN; this was higher in the extremely preterm group.

A comparison of very low birth weight preterm infants (≤1500 g) with infants heavier than 1500g detailed in Table 5 shows that there were significant differences in the maximum protein and lipid intake; the intakes were higher in the group of preterm infants with BW≤1500 g in both cases. There were also significant differences in the initial carbohydrate intake, which was higher in the case of preterm infants with BW >1500 g. On the other hand, no significant differences in the initial intake of proteins and lipids were found. The total duration in days of PN was higher in the group of preterm infants with BW ≤ 1500 g.

There were statistical and negative correlations between number of day with PN and GA (r_s_ = −0.513) and BW (r_s_ = −0.514) There was a moderate relationship, with the total number of days of PN being greater for lower GA and lower BW. 

## 4. Discussion

This study aimed to evaluate clinical practice in the neonatal unit of a tertiary hospital regarding the prescription of PN in preterm infants. The findings show that medical prescriptions conform to the protocol established in that hospital. In general, they also conform to the recommendations set forth in the guidelines of scientific societies (ASPEN, ESPEN and SENPE), although there are small differences in the guidelines for recommended macronutrient intake; for example, in the ESPEN guidelines, the recommended initiation of carbohydrates differs from that established in the ASPEN and SENPE guidelines.

Prematurity is associated with growth delay, especially with delays in neurocognitive development [20,21], as well as with other possible related consequence [22]. Many cases present restriction of postnatal growth below the 10th percentile [23], and this restriction is inversely proportional to gestational age [6]. This is due, in large part, to the nutritional deficit that develops from insufficient administration of nutrient [24]. In addition, the limited reserves of nutrients possessed by these infants cause a compromised nutritional status, making the use of PN [25] fundamental in preventing the catabolism that would otherwise occur when the continuous supply of placental nutrients ceases [26]. Therefore, from birth, adequate nutritional intake must be provided to preterm infants to meet foetal growth requirements [27,28].

Currently, in contrast to the classic administration of glucose during the first days of life, the concept of “aggressive nutrition”, which involves the early administration of lipids and a significant supply of protein along with high glucose from the first hours of life, has emerged [29]. In the case of newborns with BW<1500 g, the most recent studies show that greater protein and lipid intake (2.4 and 2g/kg/day, respectively) from the first day is safe and that it improves the infants’ nitrogen balance provided that the appropriate ratio of 25–40kcal non-protein/g protein is used [30]. Higher protein intake (3–4g/kg/day together with lipids) has not been shown to improve nitrogen balance and is also associated with increased plasma urea [30,31] and not associated with improvement in growth or neurological development [32]. Our data show that there are no differences in the initial protein and lipid intake according to GA and BW; therefore, it would be advisable to evaluate whether the use of higher intake levels in the most preterm infants should be recommended.

Regarding the intake of the micronutrients studied, our results also conform to the recommendations established in the hospital protocol; it is noteworthy that the initiation of trace elements on day 4 or 5 shows a compliance of 64.9%. This may be due to an inaccurate control of days of PN, because 86.7% of preterm infants for whom trace elements were not initiated on the established days began receiving trace elements on day 6. It should be noted that the published guidelines of the ASPEN, SENPE and ESPEN scientific societies do not specify when administration of these micronutrients should begin; this should be reviewed, because a consensus that specifies how these micronutrients should be supplied is needed [33].

In a general sense, there are studies that reflect the importance of adequate micronutrient intake in preterm infants and show that it is important to ensure their administration because some problems in growth have been related to an insufficient supply of these nutrients [34]. The administration of calcium from the first day of life is advised and is necessary to avoid the characteristic hypocalcaemia that often develops in preterm infants [35]. The trace elements, in particular zinc, which is a necessary cofactor for numerous important enzymes, must be supplemented in PN from birth [36,37,38]. In addition, insufficient vitamin intake should be avoided because it may also be involved in the development of disorders such as bronchopulmonary dysplasia, osteopaenia and intraventricular haemorrhage, among others [36].

On the other hand, the administration of PN should be accompanied whenever possible by the concomitant use of the enteral route, even if it is trophic. Maintenance of trophism improves intestinal maturation and provides other benefits such as improving postnatal growth, cardiac and respiratory diseases and reducing the risk of infections [39]. It must be taken into account that a long duration of PN can lead to complications, mainly infections [40] or metabolic complications (hyperlipidaemias, hyperbilirubinaemia, cholestasis) [41]. Therefore, PN should be terminated when it is possible to orally administer a sufficient amount of enteral nutrition (EN) to satisfy the minimum nutritional requirements equivalent to those achieved during foetal development [42]. In this study, data on EN prescriptions could not be collected; however, as expected, longer duration of PN was associated with lower GA and lower BW. This corroborates the finding that enteral tolerance in the most preterm births is achieved with great difficulty due to the infants’ gastrointestinal immaturity [24].

As for the assessment of how nutritional practice is currently being conducted in neonatal intensive care units, studies are scarce. For the most part, such studies gather information through questionnaires that are sent to specialists in neonatal intensive care, such as neonatologists, nurses and neonatal dietitians [14,17,18]. There is a risk that the data obtained in this way do not reflect the actual practice but are instead based on what would be done in theory.

Some studies have analysed real data on the prescription of supplements in preterm infants in clinical practice [15,16,17]. In this line, one study evaluated energy and macronutrient intake by the parenteral route and/or from enriched breast milk during the first postnatal week, with data collection until day 8 (DoMINO clinical trial) [15]. The authors emphasized the importance of adequate intake not only during the first days of life but beyond; they observed that although the recommended values were reached on days 6 and 7.34% and 71% of very low birth weight preterm infants did not meet the established protein and energy requirements, respectively, on day 8 [15]. According to another study conducted in the United Kingdom in which 264 cases of neonatal PN were reviewed, there was recognition of the importance of deficiency of the correct administration of PN at the care level; in 37% of the cases, the initial prescription of PN was inadequate, and in 19%, follow-up was inappropriate [16].

In another worldwide study in which 52 countries participated, a questionnaire on the nutritional strategies developed in each paediatric intensive care unit (PICU) was distributed, and the results were compared with real data from clinical practice. Differences were found that reflected an overestimation of the ability to adequately feed neonates. The results indicated that only 55% of PICUs initiated PN within 48 h of birth in patients with intolerance to EN. In general, this would be used in 72% of PICUs if EN intake did not comply with at least 50% of the caloric targets. It is important to emphasize that only 52% of these centres had an established nutritional management protocol [17]. In our study, the high compliance with the protocol by the medical staff shows that the development of nutritional protocols in PICUs is a good tool that helps ensure proper nutritional monitoring of preterm newborns [43].

Regarding the limitations of this study, it is noteworthy that only PN data were collected and that the supply of EN/breastmilk that patients received is unknown, although the enteral supply of nutrients does not contribute to the analysed data, in this study total caloric intake is not analysed, the macronutrient intake analysed refers to the minimum initial and maximum intake reached, so the supply of EN/breast milk does not influence. In addition, pathologies and/or complications that may have developed in preterm infants were not identified because the hospital protocol did not establish intake based on them, and these data were not documented in the programmes used. Finally, it must be taken into account that due to the low prevalence of preterm infants who also require PN, a total number of 155 preterm infants (over a period of 3 years) was obtained; when these infants are divided into groups, the sample number is reduced significantly in some cases, making it impossible to extrapolate the data. This limitation also makes it impossible to analyse other health outcomes such as survival rates compared compliance with the hospital protocol. This demonstrates the need for larger and multicentre studies that can produce results with greater statistical power.

This study concludes that there is a high percentage of compliance with both the nutritional protocol established in the neonatal care unit of a tertiary hospital and its suitability as judged by comparison with the recommendations of the scientific societies ASPEN, ESPEN and SENPE. On the other hand, our findings show that lower GA and BW are related to longer duration of PN and that the initial protein and lipid intake prescribed do not differ in a way that is related to these variables. In addition, the results of this study demonstrate the importance of preparing and evaluating protocols in clinical practice and the need to adapt the PN of preterm neonates to clinical protocols. For this reason, more studies analysing the management of PN in the preterm population are needed, given the scarce current evidence supplied in national and international studies. Future research should focus on establishing a consensus regarding the intake requirements in this population group. This would also make it possible to obtain more comprehensive and detailed information on the specific intake of micronutrients and trace elements.

## Figures and Tables

**Table 1 ijerph-17-02131-t001:** Sociodemographic variables and maternal history.

	Extremely Preterm	Preterm	Late Preterm	*p*-value
	(<28 Weeks)	(28–34 Weeks)	(34–37 Weeks)	
	(n = 37)	(n = 98)	(n = 20)	
Sex				*p* = 0.550 *
Male	45.9 (17))	56.1 (55)	50.0 (10)	
Female	54.1 (20)	43.9 (43)	50.0 (10)	
(%(n))				
GA (weeks)	26.3 (±1.3)	30.6 (±1.7)	34.7 (±0.7)	
mean(±SD)				
Birth weight (grams)	905.0 (±351.0)	1300.0 (±423.0)	1930.0 (±1000.0)	
median(±IQR)				
Maternal age (years)	31.0 (±11.0)	31.5 (±8.0)	34.0 (±7.0)	*p* = 0.137 **
median(±IR)				
Pregnancy	81.1 (30)			*p* = 0.110 *
Simple	18.9 (7)	62.2 (61)	70.0 (14)	
Multiple		37.8 (37)	30.0 (6)	
(%(n))				
Gestation				*p* = 0.351 *
1 gestation	23.5 (8)	37.1 (36)	35.0 (7)	
>1 gestation	76.5 (26)	62.9 (61)	65.0 (13)	
(%(n))				
Previous abortions				*p* = 0.173 *
Yes				
No	61.8 (21)	43.3 (42)	45.0 (9)	
(%(n))	38.2 (13)	56.7 (55)	55.0 (11)	
Live children				*p* = 0.829 *
1 Live	44.1 (15)	38.1 (37)	40.0 (8)	
>1 Live	55.9 (19)	61.9 (60)	60.0 (12)	
(%(n))				

Data are expressed as % (n) or means (±SD) or medians (± IQR), as appropriate. No differences between groups according to the different variables. * X^2^ Test ** Kruskal–Wallis Test GA: gestacional age.

**Table 2 ijerph-17-02131-t002:** Distribution of prescribed intakes.

Initiation PN Postnatal day 1 (% (n))	84 (130) Yes 16 (25) No
Initiation proteins (g/kg/day) (median ± IQR)	2.0 (±0.5)
Maximum proteins (g/kg/day) (median ± IQR)	3.0 (±0.2)
Initiation lipids (g/kg/day) (median ± IQR)	1.0 (±0.5)
Maximum lipids (g/kg/day) (median ± IQR)	3.0 (±0.5)
Initiation CHO (g/kg/day) (median ± IQR)	7.0 (±0.5)
Maximum CHO (g/kg/day) (median ± IQR)	11.0 (±2.5)
Initiation calcium first day (% (n))	99.4 (154) Yes 0.6 (1) No
Initiation zinc first day (% (n))	99.4 (154) Yes 0.6 (1) No
Initiation trace elements 4th-5th day (% (n))	64.9 (85) Yes 35.1 (46) No
Initiation vitamins first day (% (n))	97.4 (151) Yes 2.6 (4) No

Data are expressed as % (n) or medians (±IQR), as appropriate PN: parenteral nutrition; CHO: carbohydrates.

**Table 3 ijerph-17-02131-t003:** Compliance with the guidelines published by SENPE, ESPEN and ASPEN and with the established hospital protocol.

	SENPE 2017	*Compliance* *(%(n))*	ESPEN 2005	*Compliance* *(%(n))*	ASPEN 2002	*Compliance* *(%(n))*	Hospital Protocol	*Compliance* *(%(n))*
**Initiation PN**	Postnatal day 1	84 (130) Yes	Postnatal day 1	84 (130) Yes	Postnatal day 1	84 (130) Yes	Postnatal day 1	84 (130) Yes
		16 (25) No		16 (25) No		16 (25) No		16 (25) No
**Initiation minimum CHO**	8.6	8.3 (13) Yes	5.8	99.4 (154) Yes	8.6	8.3 (13) Yes	6	98.7 (153) Yes
**(g/kg/day)**		91.7 (142) No		0.6 (1) No		91.7 (142) No		1.3 (2) No

**Maximum CHO**	17.3	99.4 (154) Yes	12	82.1 (127) Yes	18.7	100 (155) Yes	18	100 (155) Yes
**(g/kg/day)**		0.6 (1) No		17.9 (28) No		0 (0) No		0 (0) No
**Initiation minimum lipids**	0.5	98.1 (152) Yes	0.25	98.1 (152) Yes	0.5	98.1 (152) Yes	0.5	98.1 (152) Yes
**(g/kg/day)**		1.9 (3) No		1.9 (3) No		1.9 (3) No		1.9 (3) No

**Maximum lipids**	4	100 (155) Yes	4	100 (155) Yes	3	98.1 (152) Yes	3	98.1 (152) Yes
**(g/kg/day)**		0 (0) No		0 (0) No		1.9 (3) No		1.9 (3) No
**Initiation minimum proteins**	1.5	95.5 (148) Yes	1.5	95.5 (148) Yes	1	100 (155) Yes	1.5	95.5 (148) Yes
**(g/kg/day)**		4.5 (7) No		4.5 (7) No		0 (0) No		4.5 (7) No

**Maximum proteins**	4	100 (155) Yes	4	100 (155) Yes	3.85	100 (155) Yes	4	100 (155) Yes
**(g/kg/day)**		0 (0) No		0 (0) No		0 (0) No		0 (0) No
**Initiation day calcium**	-		-		-		Postnatal day 1	99.4 (154) Yes
								0.6 (1) No
**Initiation day zinc**	-		-		-		Postnatal day 1	99.4 (154) Yes
								0.6 (1) No
**Initiation day trace elements**	-		-		-		Postnatal day 4–5	64.9 (85) Yes
								35.1 (46) No
**Initiation day vitamins**	-		Postnatal day 1	97.4 (151) Yes	-		Postnatal day 4–5	97.4 (151) Yes
				2.6 (4) No				2.6 (4) No

The compliance is shown %(n) for each data, yes (there is compliance), no (there is not compliance) PN: parenteral nutrition; CHO: carbohydrates.

**Table 4 ijerph-17-02131-t004:** Distribution of prescribed macronutrient intake according to gestational age.

Macronutrient Intake and Total Days PN	Extremely Preterm (<28 Weeks) (n = 37) (Median ± IQR)	Preterm (28–34 Weeks) (n = 98) (Median ± IQR)	Late Preterm (34–37 Weeks) (n = 20) (Median ± IQR)	*p*-Value
Initiation proteins	2.0 (±0.5)	2.0 (±0.5)	2.0 (±1.0)	*p* = 0.159 *
(g/kg/day)				
Maximum proteins	3.0 (±0.25)	3.0 (±0.2)	2.9 (±0.5)	*p* = 0.086 *
(g/kg/day)				
Initiation lipids	1.0 (±0.5)	1.0 (±0.5)	1.25 (±0.95)	*p* = 0.141 *
(g/kg/day)				
Maximum lipids	3.0 (±0.5)	3.0 (±0.5)	2.5 (±1.0)	*p* = 0.003 *
(g/kg/day)				
Initiation CHO	6.5 (±0.75)	7.0 (±0.5)	8.0 (±2.38)	*p* < 0.001 *
(g/kg/day)				
Maximum CHO	11.0 (±3.0)	11.0 (±2.0)	10.75 (±2.63)	*p* = 0.823 *
(g/kg/day)				
Total days PN	14.0 (±10.0)	8.0 (±6.0)	5.0 (±3.0)	*p* < 0.001 *
(days)				

* Kruskal–Wallis test, CHO: carbohydrates; PN: parenteral nutrition.

**Table 5 ijerph-17-02131-t005:** Distribution of prescribed macronutrient intake according to birth weight.

Macronutrient Intake and Total Days PN	BW ≤ 1500g (n = 117) *(Median ± IQR)*	BW >1500 g (n = 38) *(Median ± IQR)*	*p*-Value
Initiation proteins	2.0 (±0.5)	2.0 (±0.5)	*p* = 0.068 *
(g/kg/day)			
Maximum proteins	3.0 (±0.2)	3.0 (±0.5)	*p* = 0.002 *
(g/kg/day)			
Initiation lipids	1.0 (±0.5)	1.0 (±0.5)	*p* = 0.838 *
(g/kg/day)			
Maximum lipids	3.0 (±0.5)	2.5 (±1.0)	*p* < 0.001 *
(g/kg/day)			
Initiation CHO	7.0 (±0.5)	7.0 (±1.38)	*p* = 0.003 *
(g/kg/day)			
Maximum CHO	11.0 (±2.13)	10.25 (±2.13)	*p* = 0.130 *
(g/kg/day)			
Total days PN	10.0 (±9.0)	5.0 (±2.0)	*p* < 0.001*
(days)			

* Mann–Whitney U tests, BW: birth weight; CHO: carbohydrates; PN: parenteral nutrition.

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
