# Peer review of "Adequacy of Parenteral Nutrition in Preterm Infants According to Current Recommendations: A Study in A Spanish Hospital"

_ijerph, 2020, doi:10.3390/ijerph17062131_

Round 1

Reviewer 1 Report

Thanks for an informative piece. Below are comments for your consideration.

  1. The conclusion in the abstract is too general. I suggest you state first the key finding that "a lower GA and BW are related to a longer duration of PN" then follow with the importance of developing, standardizing, and evaluating clinical protocols. 
  2. line 112: consider using "child" in place of "son" to include both gender
  3. In addition to the stated limitation, one other limitation is the lack of outcome data. Since the study was carried out for 3 years, it might have been possible to report on the survival rates and attainment of milestones etc. among the compliance against the non-compliance groups If this data is not available then it will be helpful to state it as a limitation. 

Author Response

The changes made to the manuscript based on the reviewers' recommendations are detailed in this cover letter.  We really appreciate your positive considerations to the manuscript: “Adequacy of parenteral nutrition in preterm infants according to
current recommendations: a study in a Spanish hospital”
ID: ijerph-742454. We appreciate the time and efforts by the editor and referee in reviewing this manuscript and for the useful constructive comments. We have addressed all issues indicated in the review report and believed that the revised version has hopefully improved.

All authors of this article appreciate your trust in the article and your comments to improve its publication. We thank the reviewers’ suggestions.

Please note that we have written the answers in the text in supplied Word format.

Response:

1.We highlight in the conclusions section of the abstract the key finding that a lower gestational age and birth weight are related to a longer duration of parenteral nutrition. Conclusions: A lower gestational age and birth weight are related to a longer duration of parenteral nutrition. The results of this study demonstrate the importance of developing and evaluating protocols in clinical practice.

2. In line 114 in Word format (reviewer 1, line 112) We write "child" in place of "son" to include both gender.

3. We add the limitation that due to the sample number other health outcomes such as survival rates compared compliance with the hospital protocol cannot be evaluated. This limitation also makes it impossible to analyze other health outcomes such as survival rates compared compliance with the hospital protocol”.

Reviewer 2 Report

I would to thank the authors to report their data. Regarding the review process, please find attached my kindly suggestions to be consider. 

Author Response

Dear Hazel Liu

The changes made to the manuscript based on the reviewers' recommendations are detailed in this cover letter.  We really appreciate your positive considerations to the manuscript: “Adequacy of parenteral nutrition in preterm infants according to
current recommendations: a study in a Spanish hospital”
ID: ijerph-742454. We appreciate the time and efforts by the editor and referee in reviewing this manuscript and for the useful constructive comments. We have addressed all issues indicated in the review report and believed that the revised version has hopefully improved.

All authors of this article appreciate your trust in the article and your comments to improve its publication. We thank the reviewers’ suggestions.

Please note that we have written the answers in the text in supplied Word format.

Reviewer 1

  1. The conclusion in the abstract is too general. I suggest you state first the key finding that "a lower GA and BW are related to a longer duration of PN" then follow with the importance of developing, standardizing, and evaluating clinical protocols. 
  2. line 112: consider using "child" in place of "son" to include both gender
  3. In addition to the stated limitation, one other limitation is the lack of outcome data. Since the study was carried out for 3 years, it might have been possible to report on the survival rates and attainment of milestones etc. among the compliance against the non-compliance groups If this data is not available then it will be helpful to state it as a limitation. 

Response:

1.We highlight in the conclusions section of the abstract the key finding that a lower gestational age and birth weight are related to a longer duration of parenteral nutrition. Conclusions: A lower gestational age and birth weight are related to a longer duration of parenteral nutrition. The results of this study demonstrate the importance of developing and evaluating protocols in clinical practice.

  1. In line 114 in Word format (reviewer 1, line 112) We write "child" in place of "son" to include both gender.
  2. We add the limitation that due to the sample number other health outcomes such as survival rates compared compliance with the hospital protocol cannot be evaluated. “This limitation also makes it impossible to analyze other health outcomes such as survival rates compared compliance with the hospital protocol”.

Revisor 2

Introduction

Broad comments highlighting areas of strength and weakness.

In general, it is a structured section, clean and focus on. However, some recommendations could improve the section.

  1. Line 42 The authors mentioned the early postnatal nutritional adequacy is related to improve of intrauterine growth, which is intrauterine growing parameter. Maybe, they could explain more details this relationship to understand it.

  1. Line 45. The authors mentioned the contribution of right postnatal nutrition is associated to decrease some factors as a social cost. I would like if they could give more details of how happens this relation.

  1. The idea of lines 47-48 is repeated in the lines 51-52. The authors could summarize one of them.

Response:

Line 44 in Word format (reviewer 1, line 42): We correct "intrauterine" growth restriction for "postnatal" growth restriction. This was a mistake in the writing.

Line 47 in Word format (reviewer 1, line 45) We give more details of how happens the relation between the correct intervention postnatal nutritional and the healthcare costs.“since poor growth in children is clearly associated with the risk of developing physical or neurological problems that require greater health and social care, leading to a higher costs.”

The writing the idea lines 51-52.has been modified.

To ensure proper nutritional intervention, it is essential that neonatal units develop protocols should be based on recommendations that have been validated and agreed upon by scientific societies.

 Materials and Methods

  1. Procedure subheading. How the nutritional intakes (g/kg/day) were calculated? It already in the hospital software?
  2. The acronym for interquartile range is usually IQR, if the authors could change in whole manuscript could be great.
  3. What tests were used when the variables were normal?

Response:

  1. We clarify how nutrients intake are calculated“The pharmacist enters the nutritional intakes prescribed by the doctor and the data of the newborn as the weight in the Hospiwin program. This program calculates the total nutrient intakes that must be prepared to administer to the newborn.”
  2. We change the acronym for interquantile range, we write IQR.
  3. In relation to the question “What tests were used when the variables were normal? No tests were performed to compare variables with normal distribution, since the only variable that was normal is gestational age. At no time several variables were compared with normal distribution.

 Results:

In general, if the authors mentioned “significantly different” or “statistically differences” it is not necessary mention the p-value in the text. The tables should specify in the footnotes not only the acronyms but also how the data shown and what test was used. Please, re-check the tense in whole section, describe the data in present or past but always to be consistent.

  1. Table 1. The “total” column is not necessary, and maybe the authors could clean it if they report these data in the text.
  2. Table 2. The data could be more interested if the authors spread the table according to the infants with and without PN at day 1 and report the stat.
  3. “Initiation PN first day” could write as “initiation PN at postnatal day 1”.
  4. The table 3 is really confused and difficult to understand. Maybe the authors could explain what is the proposal of the table in the text and re-do it and clean up. The association acronyms in the footnotes is also important.
  5. The data in table 4 and 5 are expected. Maybe the authors could say it in the text.
  6. I suggest the description of correlation would be as “…there were statistical and negative correlations between number of day with PN and GA (rho=-0.513) and BW (rho=-0.514)…”
  7. Discussion:in general, is a nice section, very clear and reviewed. Some suggestion to be considered by the authors, if they could emphasize their data and establish what is the relevant of this study in the European union or Spanish territory, could be extrapolated to another country?

  1. Lines 169 to 175. I want to thank the authors; it is focus paragraph and pretty synthetic.

Response:

We appreciate your suggestions, and will try to respond to each of them:

  • We re-check the results and we describe the data in past.
  • We cleaned up p-value in the text because we mention “significantly different” or “statistically differences” in the text.
  • In addition, we specify in the footnotes of tables how the data shown and what test was used.

  1. In Table 1, we clean the “total” column because we report these results in the text.

  1. 16% of neonates did not start parenteral nutrition on the first postnatal day. We think that comparing 84% of the sample against 16% would not have reliable results.

  1. Thanks for this acute observation. We have modified and written "Initiation NP on the first postnatal day".

  1. The table 3. We have put the table in a horizontal position for a better understanding and comprehension, hoping that this will be enough.

  1. In relation to the recommendation “The data in table 4 and 5 are expected. Maybe the authors could say it in the text” We think that these comments should be commented in the discussion of manuscript, comparing with other previous studies. We have done this.

  1. We change the description of the correlation on lines 169 and 170.

  1. Some suggestion to be considered by the authors, if they could emphasize their data and establish what is the relevant of this study in the European union or Spanish territory, could be extrapolated to another country?

This observation may be reflected in the conclusions we have indicated from the study, in the following paragraph In addition, the results of this study demonstrate the importance of preparing and evaluating protocols in clinical practice and the need to adapt the PN of preterm neonates to clinical protocols. For this reason, more studies analysing the management of PN in the preterm population are needed, given the scarce current evidence supplied in national and international studies. Future research should focus on establishing a consensus regarding the intake requirements in this population group. This would also make it possible to obtain more comprehensive and detailed information on the specific intake of micronutrients and trace elements”.

Specific comments referring to line numbers, tables or figures.

Response:

  1. Double-check the abstract section. What the author means with “meet foetal growth”? The correlation statement could be improving as “The GA and BW were negative correlated with long time of PN….”.

  • We change “meet” for “accomplish” in abstract line 14.

  1. Line 40. Change “younger GA” for “low GA”.

  • Line 42 in the Word format, we change for “low GA”.

  1. Line 87 and 89. The manufacture of Mizar and HospiWin could be useful to report.

  1. Line 100. “…when they do not.” Is colloquial speech, please try to found other expression.

  • We change the expresión en the line 107 in Word format “when the values not follow a normal distribution”.

  1. Line 111. “…approximately half (47.7%)…” same comments than before.

  • Has been modified by “A total of 66.2% of the mothers had been pregnant before and the 47.7%” in the line 118 Word format.

  1. Line 112. Change “living son” for “lived labor”.

  • In the line 119 we've changed for“lived child”.

  1. Line 146. “statistically significant” is redundant.
  • The change has been made.

  1. Line 147. Substitute “among the” for “within groups”.

  • In the line 152 is changed to “within groups”.

  1. Line 155. Say “…there are significant differences…”and in the lines 157 and 159 also.

  • We say “significant differences” lines 160, 162, 163.

 10 - 11. Line 156. Intakes.

 There are some type font and size errors. Please re-check format. Subheading 2.1.1, table 4 (font size), line 111 (double “had” word), line 135 to 136 (space).

  • We correct it.

10-11-12. Reference 5. The available is missing.

       Line 190. The reference 30 is bad typed.

       Line 247. The reference 17 is bad typed.

  • We correct it.

Thank you very much for your consideration.
